# Position: Hippocampal Explicit Memory Is the Cornerstone for AGI

**Sangjun Park** [1] [2]

## Abstract

Large Language Models (LLMs) have demonstrated remarkable capabilities across various tasks, raising expectations for Artificial General Intelligence (AGI). This position paper argues that integrating explicit memory is the cornerstone for advancing LLMs toward AGI. The key reason is that the underlying learning mechanism of LLMs is highly analogous to human implicit memory. However, higher-order cognitive functions necessary for AGI, such as long-term strategic planning, metacognition, and symbolic reasoning, heavily rely on hippocampal explicit memory and cannot arise solely from implicit statistical learning. Drawing on findings from neuroscience, I advance this perspective and complement it with computational requirements for artificial explicit memory systems, hoping to foster further research and lay the groundwork for explicit memory integration.

## 1. Introduction

Large Language Models (LLMs) have recently achieved remarkable success in the field of natural language processing (Comanici et al., 2025; Singh et al., 2026), opening new possibilities in artificial intelligence (Grattafiori et al., 2024; Yang et al., 2025; DeepSeek-AI et al., 2025). Trained on vast amounts of text data, these models comprehend context and demonstrate near-human performance in tasks such as writing, question-answering, code generation, and conversational assistance (Wei et al., 2022; Srivastava et al., 2023; Liang et al., 2023). Notably, advanced models like GPT-5, utilizing hundreds of billions of parameters, can solve complex problems and have practical applications across various domains (He et al., 2025; Chakrabarty et al., 2022; Colombo et al., 2024). Some researchers even argue that LLMs can be considered the initial stages of Artificial General Intelligence (AGI), due to their ability to understand and address intricate issues (Bubeck et al., 2023).

However, despite their impressive performance, LLMs still face significant challenges such as hallucination, difficulties in planning, and limitations in logical reasoning (Gendron et al., 2024; Yao et al., 2023; Tian et al., 2024; Song et al., 2025; Valmeekam et al., 2022). Since LLMs cannot store and use information dynamically over long periods, memory has always been considered a notable weakness (Goertzel, 2023; Feng et al., 2024; Zhong et al., 2024; Morris et al., 2024). I regard memory as the key point for addressing these issues and also for developing higher-order capabilities essential for AGI (Collins et al., 2024), such as dynamic learning, reflection, and metacognition. While there are various discussions regarding the definition of AGI, I consider Human-Level AI to be the defining criterion for AGI: an AI system with general, human-level ability to learn, reason, and apply knowledge across all cognitive tasks and domains. Although memory is commonly perceived as merely information storage, in reality, it is an integral component of all learning processes in humans (Atkinson & Shiffrin, 1968; Anderson, 1982). Constraints on memory are directly linked to limitations in cognitive ability, highlighting its pivotal role in the evolution of artificial intelligence.

In this paper, I argue that the fundamental learning mechanism of LLMs can be compared to learning based on implicit memory in biological systems. I also suggest that the cognitive functions of LLMs are restricted to implicit learning. Based on this perspective, **I claim that realizing AGI should involve the integration of explicit memory.**

To support this claim, the remainder of this paper is organized as follows. Section 2 reviews the fundamental concepts and mechanisms of human explicit and implicit memory systems. Section 3 analyzes the underlying learning processes of current LLMs, illustrating their structural and functional resemblance to implicit memory. Section 4 explores the distinct cognitive functions supported by each memory system, to clarify the operational boundaries of each memory. Building upon these insights, Section 5 details the specific limitations of current LLMs and highlights why explicit memory is essential for higher-order cognition. Section 6 then formally defines the computational requirements for developing an artificial explicit memory system.

---

[1]Department of Computer Science, University of Texas at Austin, TX, USA [2]Cognizant AI Labs, San Francisco, CA, USA. Correspondence to: Sangjun Park <sangjun@cs.utexas.edu>.

*Proceedings of the 43rd International Conference on Machine Learning*, Seoul, South Korea. PMLR 306, 2026. Copyright 2026 by the author(s).

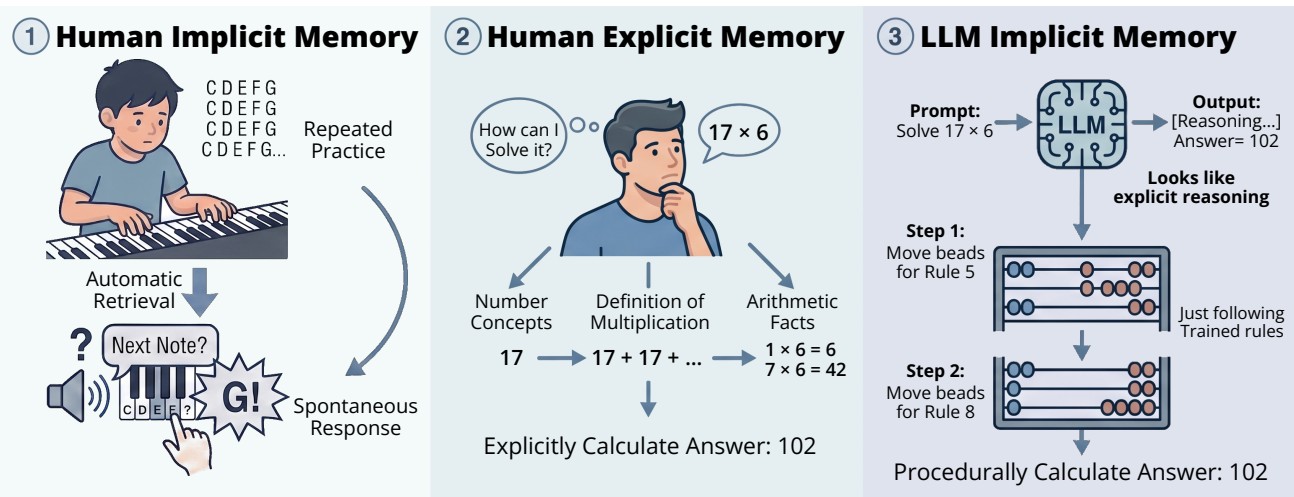

*Figure 1.* Comparison of three memory and learning paradigms. **(1)** Human implicit memory forms through repeated practice and produces automatic, spontaneous responses (e.g., predicting the next note in a rehearsed melody). **(2)** Human explicit memory supports conscious, knowledge-based problem solving: $17 \times 6$ is decomposed via number concepts, the definition of multiplication, and arithmetic facts to explicitly compute 102. **(3)** An LLM, given the same prompt, emits a reasoning-like trace ending in 102. Although the output looks like explicit reasoning, the underlying process is closer to operating an abacus: the correct answer can be produced automatically by just following learned manipulation rules (rule indices are illustrative), with no semantic understanding of the numbers or the procedure.

Section 7 examines alternative perspectives, Section 8 discusses broader implications, and Section 9 concludes the paper with directions for future research. In the Appendices, Appendix A reviews recent advancements in neural memory. Appendix B further discusses theoretical and practical considerations such as substrate independence and causality. Finally, Appendix C presents empirical examples demonstrating the absence of explicit memory in current LLMs.

## 2. Explicit and Implicit Memories

LLMs naturally possess memory, and understanding the memorization process in LLMs can be greatly enriched by drawing parallels with biological systems. Human memory is structured to store and utilize multiple types of information (Tulving, 1972; Baddeley & Hitch, 1974), enabling various cognitive functions. This section provides the basic concepts and operation mechanisms in each memory system. Please refer to Figure 1 for illustrative examples of explicit and implicit memory systems in humans and LLMs.

### 2.1. Basic Concepts

**Explicit Memory** Explicit memory, also known as declarative memory, handles information related to facts, events, or experiences (Tulving, 1985). Explicit memory is typically divided into two main categories: episodic memory and semantic memory. Episodic memory preserves personal experiences tied to specific times and places (Tulving, 2002), so it is closely related to autobiographical memory and is instrumental in reconstructing past experiences (Addis et al.,

2007). In contrast, semantic memory refers to the memory of general knowledge, concepts, language, and facts, which can be accessed without relying on specific experiential contexts (Patterson et al., 2007). Explicit memory is primarily associated with higher-level cognitive functions, such as active learning, problem-solving, and language comprehension. Episodic memory, in particular, is linked to executive functions like self-reflective thinking and future planning (Schacter & Addis, 2007; Schacter et al., 2017), while semantic memory is essential for knowledge-based reasoning, such as language processing and categorization (Binder & Desai, 2011). The hippocampus and medial temporal lobe are responsible for these processes (Squire, 2004).

**Implicit Memory** Implicit memory operates mainly by influencing performance and behavior through unconscious learning (Schacter, 1987). It is typically divided into procedural memory, classical conditioning, and priming. Procedural memory mainly involves motor skills and habits, such as riding a bicycle or playing a musical instrument, which are performed automatically without needing explicit recall (Doyon & Benali, 2005). These skills are developed through repeated practice and learning, allowing them to be executed effortlessly (Cohen & Squire, 1980). Classical conditioning forms stimulus-stimulus associations, while habit learning forms stimulus-response associations through extensive repetition (Clark & Squire, 1998; Graybiel, 2008). Implicit memory is a key factor in cognitive and behavioral abilities such as habituation, automated skill performance, and emotional learning. This type of memory is closely connected to brain structures like the amygdala, cerebellum, and basal ganglia (Squire & Dede, 2015).

## 2.2. Formation Mechanism

**Explicit Memory** The formation of explicit memory occurs through strong synaptic strengthening in the main neural circuits of the hippocampus and medial temporal lobe when new events are encountered (Scoville & Milner, 1957; Squire & Zola-Morgan, 1991). The hippocampus encodes events through sparse representations that index distributed cortical patterns of experience (Teyler & Rudy, 2007; Goode et al., 2020). Specifically, the entorhinal cortex (EC) serves as the gateway between the neocortex and hippocampus, relaying experience into the hippocampal trisynaptic pathway (EC → dentate gyrus → CA3 → CA1). This pathway facilitates memory encoding, where each stage exhibits distinct synaptic plasticity (Andersen et al., 1971).

First, in the dentate gyrus (DG), neural signals are sparsely encoded (Jung & McNaughton, 1993), selectively activating a small subset of neurons and thereby strongly inducing long-term potentiation (LTP). If the neurons were densely coded, a large number of neurons would be redundantly activated, leading to overlap and interference between distinct concepts and episodes. However, sparse coding ensures that even similar information is clearly separated (pattern separation), allowing distinct and non-overlapping memory representations (Yassa & Stark, 2011; Leutgeb et al., 2007).

Next, the CA3 region binds these sparsely active neurons through its recurrent collateral circuit, forming an autoassociative network. Because NMDA receptor-dependent LTP can be induced by only a few stimulations, CA3 rapidly stores the co-active elements of an episode as a single bound pattern that can later be recovered from a partial cue (Nakazawa et al., 2002; Rebola et al., 2017).

Finally, the CA1 region serves as the output stage of the hippocampus (Nakazawa et al., 2004), integrating information received from CA3 and transmitting it to the cerebral cortex and other brain areas. By regulating the hippocampal output pathways, CA1 contributes to the comprehensive management of memory, facilitating the transition of temporary hippocampal memories into long-lasting neocortical representations (Malenka & Bear, 2004).

**Implicit Memory** Implicit memory is primarily formed in subcortical structures such as the basal ganglia, cerebellum, and amygdala (Alexander et al., 1986; Conway, 2020). Among these, the basal ganglia are crucial in decision-making for behavior through the cortico-basal ganglia-thalamocortical loop (Packard & Knowlton, 2002), which repeatedly receives dense neuronal inputs from the cerebral cortex. For example, Medium Spiny Neurons (MSNs) in the striatum of the basal ganglia express D1 (direct pathway) and D2 (indirect pathway) dopamine receptors and respond sensitively to reward prediction and dopamine signals (Hikosaka & Isoda, 2010; Gerfen & Surmeier, 2011).

The main stages of learning occur as follows:

1. **Prediction**: Anticipating a reward based on performing a specific behavior in the current context.

2. **Action**: When the behavior is performed, signals from the cortex are densely transmitted to the striatal MSNs.

3. **Outcome Evaluation**: Assessing whether the behavior resulted in a reward, failure, or punishment.

4. **Synaptic Changes**: Strengthening the direct pathway if the reward is greater than expected, strengthening the indirect pathway otherwise.

Since dopamine-driven synaptic strengthening or weakening happens after behavior, a one-time result typically does not cause significant synaptic change (Schultz et al., 1997). Repeated trials are necessary to solidify the connection between a specific situation and the corresponding behavior, creating a stimulus-action mapping. Once a behavior becomes part of procedural memory or habit, it is performed or inhibited automatically when the same situation occurs, demonstrating implicit learning. Overall, implicit memory develops gradually over time, requiring numerous repetitions and trial-and-error (Montague et al., 2004).

**Comparison** Explicit memory activates a selective set of neurons (sparse coding) to create clear memories. In contrast, implicit memory relies on behavior and outcomes, gradually recruiting many neurons (dense coding) that operate simultaneously. Explicit memory forms rapidly and powerfully, whereas implicit memory develops through long-term repetition, marking a key difference between the two types of memory (Squire & Zola, 1996; Reber, 2013).

## 2.3. Retrieval Mechanism

**Explicit Memory** Explicit memory retrieval centers on the hippocampal-cortical loop, where contextual cues are essential (McClelland et al., 1995; Wiltgen et al., 2010; Kumaran et al., 2016). For example, when presented with the word "summer", this cue is transmitted through the entorhinal cortex (EC) to activate the CA3 region. The CA3 region performs pattern completion through its recurrent circuitry, which can reconstruct an entire stored memory pattern even when only a few neurons are activated (Nakazawa et al., 2002; Rolls, 2013). For instance, the word "summer" might activate neurons associated with sensory elements like the beach, sand, the sound of waves, and the smell of the breeze.

This reconstructed memory pattern is then relayed back to the broader cortex (Eichenbaum, 2000; Takashima et al., 2006). Passing through the CA1 region, the pattern interacts bidirectionally with various areas of the cerebral cortex, such as the prefrontal cortex, sensory cortex, and association cortex, synchronizing different elements of the memory (Frankland & Bontempi, 2005; Mattar & Daw, 2018).

In summary, the process of retrieving explicit memory can be outlined as follows:

1. A contextual cue or intention initiates the hippocampal-cortical loop.

2. Signals from the EC activate specific neurons in the CA3 region.

3. The CA3 region's recurrent circuitry reconstructs the entire memory pattern based on the activated neurons.

4. The reconstructed pattern is relayed through the CA1 region to the cerebral cortex, synchronizing the elements of the memory.

5. Top-down attention from the prefrontal cortex modulates the retrieval process, enabling coherent and integrated memory to emerge consciously.

This process allows rich and multi-layered memories to be replayed from a single cue or piece of information. Through this process, the hippocampus reactivates the distributed neural patterns of a past episode, effectively recreating the entire context of the original experience.

**Implicit Memory** Implicit memory retrieval happens automatically (Graybiel, 2008; Hikosaka & Isoda, 2010). When a familiar stimulus appears, stored stimulus-response patterns in the striatum are activated via MSNs (Yin & Knowlton, 2006). For example, if you suddenly need to stop while driving, your foot instinctively presses the brake. This occurs because cortical signals reach the striatum, exciting MSNs previously associated with that action (Kreitzer & Malenka, 2008). The MSNs then facilitate action selection through either the direct (D1) or indirect (D2) pathway. The direct pathway enables the action, while the indirect pathway suppresses certain actions (Tritsch & Sabatini, 2012). In short, implicit memory retrieval reactivates learned behaviors automatically in response to familiar cues.

**Comparison** Explicit and implicit memory retrieval operate through distinct mechanisms tailored to their unique functions. In explicit memory retrieval, sparse coding plays a key role in pattern completion. A small number of neurons, activated by a cue, trigger the reconstruction of the entire memory through highly selective connections. This process allows the memory system to retrieve detailed and multi-faceted experiences by filling in missing details and linking related sensory and contextual elements into a coherent whole. Implicit memory, on the other hand, operates via learned stimulus-response patterns. When a familiar stimulus is encountered, the system activates the corresponding response pattern automatically, allowing previously learned behaviors or routines to be executed efficiently. Unlike explicit memory, implicit retrieval does not reconstruct a broader memory context but instead focuses on directly accessing and applying specific learned associations.

## 3. Nature of Learning in LLMs

This section examines the learning and reasoning processes of LLMs, focusing on how their low-level mechanisms relate to explicit and implicit memory systems. In doing so, it reveals key distinctions from hippocampal explicit memory and indicates that the core characteristics of LLMs more closely resemble implicit learning and habituation.

**Gradual Learning** Explicit memory can encode new episodes vividly with just one exposure when strong stimuli are present or attention is engaged. In contrast, implicit memory relies on gradually accumulating associations between stimuli, actions, and rewards (or punishments). For instance, repeated input projection strengthens or weakens synapses incrementally, guided by dopamine signals reflecting reward prediction errors. Similarly, LLM training follows a comparable pattern. Model parameters do not change drastically with one or two examples. Instead, the model adjusts its weights cumulatively by repeatedly observing numerous examples within a vast corpus, aiming to reduce errors (loss). This gradual and repetitive learning mirrors the characteristics of implicit memory formation.

**Dense and Distributed Coding** Explicit memory is characterized by sparse coding to ensure pattern separation. In contrast, subcortical circuits receive dense signals from the cortex. LLMs exhibit a similar structure, with their network broadly interconnected. Their activations for specific contexts are not confined to isolated nodes but occur across a wide range of parameters. These features of distributed coding and large-scale parallel activations are closely aligned with the implicit processing of the basal ganglia.

**Error-Driven Learning** Implicit learning involves dopamine neurons strengthening or weakening synapses based on reward prediction errors. In contrast, learning in explicit memory relies on the associations of events or pieces of information, rather than on an error-based mechanism. The hippocampus rapidly encodes new events, combining elements presented together in time and space into a single episode. For LLMs, the loss function works in a similar way to dopamine signals, adjusting model parameters through backpropagation based on the difference between the actual output and the ground truth across large datasets. Thus, the error minimization mechanism based on prediction errors is a key shared feature of implicit memory and LLMs, distinguishing them from hippocampal learning. This distinction suggests that the underlying learning mechanism of LLMs is closely aligned with implicit learning.

**Automatic Action** Explicit memory retrieval through the hippocampus can reconstruct past episodes richly from just one contextual cue. In contrast, implicit memory retrieval in the striatum triggers automatic procedural responses without

reconstructing any past episodes. In the case of LLMs, when specific prompts (contexts) are provided, the process does not involve reconstructing an entire episode like the hippocampus. Instead, LLMs follow a procedure where their learned parameters automatically determine the optimal next word. This resembles how humans unconsciously execute habits or conditioned responses based on implicit memory, suggesting that LLM retrieval mechanisms are based on the stimulus-response mapping of the basal ganglia.

**Fundamental Similarities** In summary, the low-level learning mechanism of LLMs is closer to implicit memory systems, specifically habituation and procedural learning, than to explicit memory processes. This perspective provides valuable insights into understanding LLMs as systems that require gradual and repetitive training, and exhibit the automatic retrieval characteristics of human implicit learning.

## 4. Functional Specificity of Memory

Explicit and implicit memory serve distinct cognitive roles and are not easily interchangeable. This section explores the unique cognitive abilities supported by each memory system. By examining these differences, I indirectly reveal the cognitive functions that LLMs, which parallel implicit memory, are capable of possessing and those they cannot.

**Pattern Recognition** Pattern recognition is the ability of humans to identify specific rules or consistencies in a complex environment. For instance, people can quickly analyze facial features like eyes, nose, and mouth to identify a specific person. In language comprehension, even when spelling is incorrect or sentences are incomplete, humans can capture the meaning by recognizing patterns in words and context.

Research on statistical learning demonstrates that human pattern recognition and perception are deeply rooted in the implicit memory system. Fiser & Aslin (2002a;b) showed that infants have the ability to automatically learn statistical patterns of co-occurrences and conditional probabilities between visual objects. Turk-Browne et al. (2005) revealed that visual statistical learning is automatic and implicit, despite being gated by selective attention. Additionally, Batterink & Paller (2017) used EEG measurements to track brain activity related to implicit learning while participants detected patterns in word sequences. Ultimately, pattern recognition is an automated information processing function based on implicit memory, which is consistently supported by historical theory and modern experimental evidence.

**Language Ability** The interaction between explicit memory and implicit memory in language processing has been demonstrated through various studies, emphasizing that fluent communication heavily relies on implicit memory (Graf & Schacter, 1985). Ullman's Declarative/Procedural model (Ullman, 2004) posits that the mental lexicon, which stores word-specific knowledge, depends on explicit memory, while implicit memory plays a key role in the automatic application of grammatical rules. Research on patients with hippocampal damage (Vargha-Khadem et al., 1997; Schmolck et al., 2002) showed that even when episodic memory is impaired, semantic memory remains intact, enabling normal language abilities. This suggests that language fluency operates independently of explicit memory.

In contrast, studies revealed that dysfunctions in the basal ganglia disrupt grammatical processing and automated language use, highlighting the critical contribution of implicit memory in procedural language functions (Lieberman et al., 1992; Ullman et al., 1997). Finally, Booth et al. (2007) showed that the basal ganglia and cerebellum interact with brain regions responsible for phonological processing, supporting the refinement and automation of linguistic processes. This underscores the essential role of implicit memory in natural and flexible communication. Therefore, language ability can be understood as relying on a dual memory system, where explicit memory processes new information during the initial stages of learning, and repeated use transitions it into implicit memory for automation.

**Logical Reasoning** Mathematical and logical reasoning rely heavily on semantic memory, a subset of explicit memory, as evidenced by various neuroscience studies. Friedrich & Friederici (2013) found that the interaction between the prefrontal cortex and hippocampal systems is essential for processing the semantics of mathematical logic. Menon (2016) further explained that declarative memory is an integral part of forming associative memories, which allows for generalization beyond superficial problem attributes. Additionally, research by Evans et al. (2024) showed that declarative memory is a key predictor of elementary school students' mathematical skills, whereas procedural memory has little influence. Although Fayol & Thevenot (2012) observed that simple operations like addition and subtraction can be performed through procedural memory, the converging evidence suggests that complex mathematical and logical reasoning critically depends on explicit memory.

**Executive Function** Executive functions are key components of higher-order cognitive abilities in humans. They encompass various subdomains that help achieve goals, solve problems, and adapt to changing environments. In particular, cognitive flexibility, planning, decision-making, and task-switching are critical skills that enable individuals to adapt to changes, achieve goals systematically, make optimal decisions, and efficiently transition between tasks. Explicit memory is a crucial factor in executive functions through the retrieval of past experiences, which supports complex cognitive processes such as planning and decision-making.

Klein et al. (2002) explained that explicit memory provides contextually relevant information to decision-making sys-

tems, enabling more accurate judgments. This role of memory is also demonstrated in multitasking studies by Burgess et al. (2000), who found that retrospective memory supports prospective memory and planning abilities, while damage to related brain regions can impair these functions. Hassabis & Maguire (2007) emphasized that the scene construction process of episodic memory is crucial for future-oriented thinking and goal-directed actions. Whittington et al. (2020) computationally demonstrated that the hippocampal formation organizes knowledge into structured relational maps.

On the other hand, deficits in explicit memory are closely linked to impaired executive functions. For example, Johns et al. (2012) reported that individuals with Down syndrome who performed poorly on explicit memory tasks showed weaknesses in executive functions such as working memory and cognitive flexibility. Lastly, Pedraza et al. (2024) demonstrated that implicit statistical learning can compete with executive functions and even negatively impact declarative learning processes, such as goal-directed behaviors. In summary, explicit memory provides the foundation for executive functions by enabling the retrieval and manipulation of information. It is essential for higher-order processes like planning, judgment, and future-oriented thinking.

**Metacognition** Explicit processes are central to metacognition and reflection, enabling individuals to monitor and regulate their cognitive behaviors. The feeling of knowing (FOK) exemplifies this, as explored by Irak et al. (2019), who linked FOK judgments to specific neural components, emphasizing the role of explicit mechanisms in predicting memory retrieval. Nelson (1990) similarly highlighted that FOK relies on explicit control and monitoring processes to resolve the paradox of predicting recognition of inaccessible items. Fleming & Dolan (2012) further tied metacognitive accuracy to the prefrontal cortex, showing that reflective judgments depend on explicit representations. This aligns with Klein & Gangi (2010)'s view of the self as a system of interrelated explicit processes, crucial for integrating episodic memory and semantic knowledge. In nonhuman primates, Hampton et al. (2020) provided evidence of explicit memory systems supporting metacognition, reinforcing their evolutionary significance. These studies illustrate that explicit mechanisms are essential for reflective thought, enabling self-awareness and adaptive decision-making.

**Mental Simulation** Episodic memory, as a constructive and flexible system, plays a critical role in mental simulation by enabling the recombination of past experiences to envision future scenarios (Schacter & Addis, 2007). This capacity for "mental time travel" is uniquely human and vital for adaptive planning (Suddendorf & Corballis, 2007). Episodic memory and simulation share the default mode network, which supports scene construction and autobiographical thought, dividing into distinct subsystems handling memory retrieval

and reflective social cognition (Hassabis & Maguire, 2007; Andrews-Hanna et al., 2014). Moreover, while episodic and semantic details interact dynamically, episodic simulation often demands greater cognitive control to construct vivid hypothetical scenarios (Benoit & Schacter, 2015; Devitt et al., 2017). Together, these findings highlight how episodic memory underpins the mental simulations that enable humans to anticipate and navigate the future.

In conclusion, cognitive functions are closely linked to specific memory systems, and understanding these dynamics allows us to better identify the strengths and weaknesses of LLMs. Due to their exceptional performance in implicit memory-based processes, LLMs excel at recognizing patterns, generating fluent language, and leveraging statistical associations within large datasets. However, their various limitations become apparent in tasks requiring explicit memory, which will be discussed in detail in the next section.

## 5. Need for Explicit Memory

The previous section examined how explicit and implicit memory systems influence distinct cognitive functions. This section addresses the key limitations that LLMs must overcome to evolve into AGI. It is notable that these limitations closely align with the inherent constraints of implicit memory, which lacks mechanisms for rapid and context-rich learning. Explicit memory, in contrast, offers a potential pathway to overcoming these challenges by enabling higher-order cognitive abilities of humans. Refer to Appendix C for actual examples showing these weaknesses.

### 5.1. Higher-Order Learning

Humans utilize a wider range of higher-order learning mechanisms based on explicit memory. First, humans are capable of **dynamic learning**, continuously acquiring new knowledge over time. In contrast, the learning process in LLMs is strictly confined to the training stage, which limits their potential to function as real-time learners. **One-shot learning** is especially crucial for immediate adaptation of agents in settings such as narrative memory, personalized contexts, or unpredictable situations. This instant learning is also enabled by explicit memory, through substantial synaptic weight adjustments after just a single experience.

Furthermore, **episodic learning**, which combines knowledge with its source and spatiotemporal context, forms the foundation of the "feeling of knowing" and metacognition. This type of learning grants individuals their unique self-narratives. Humans can also learn through semantic memory, which allows the understanding of abstract relationships found in books without requiring direct experience or links to rewards. This capacity for **abstract learning** is a distinctive ability that sets humans apart from other primates.

Finally, human explicit memory systems are built on associations between knowledge structures. When a new concept is introduced, it is not treated as an isolated sample. Instead, it integrates into existing semantics by forming associations with them, enabling **continual learning** by updating or expanding knowledge in a natural way. This higher-order learning is essential for bridging the gap between LLMs and AGI and strongly supports the need for explicit memory.

## 5.2. Metacognition

**Hallucination** has been a well-known and persistent issue in LLMs since their early stages. In essence, hallucination can be attributed to a lack of **metacognition**. Metacognition entails the ability to distinguish between what one knows and does not know, as well as to keep track of the source and context of one's knowledge. Since LLMs lack episodic metadata about their knowledge and learn by forming probabilistic distributions based on data, the occurrence of hallucinations is an expected byproduct.

The problem of **consistency**, where LLMs provide different answers to the same question under subtle noise, is also inevitable due to the absence of metacognition and any grounding for the knowledge. In contrast, humans can store their actions as memories in specific situations, reflect on the outcomes later, and engage in **mental simulations** or **reflections**. This ability relies on episodic memory, which is essential for cognitive abilities related to metacognition.

## 5.3. Logical Reasoning

As mentioned earlier, simple arithmetic is often automatically mapped into human procedural memory through repeated practice. However, when **arithmetic problems** become more complex and involve longer numbers, humans can solve them given enough time, while LLMs tend to struggle. Logical reasoning, such as **mathematical reasoning, causal reasoning, or deductive reasoning**, heavily relies on concepts stored in semantic memory and the associations between them. The difficulty of logical reasoning lies in the fact that the core logical concepts and their relationships are so abstract that the possible variations of corresponding low-level representations can be practically infinite. For example, the principle of multiplication is simple, but the specific expressions representing multiplication equations are infinite. Such representational diversity makes statistical learning difficult. Solving such high-level logical problems using only statistical habituation, without relying on semantic memory, would be nearly impossible.

## 5.4. Executive Function

Executive functions encompass a variety of sub-cognitive functions. Among these, regulatory abilities like **inhibition**

and **task-switching** are not prominently displayed in current LLMs, as they are not necessary without episodic memory. **Planning**, while long considered a weakness of LLMs, has seen significant improvement with repeated efforts and the development of models specialized in reasoning. Executive functions heavily rely on short-term storage, often referred to as working memory. Within the context window that corresponds to working memory, LLMs demonstrate strong reasoning capabilities, allowing for planning within a limited scope. However, to reach human-level capabilities, such as writing a 1,000-page book or completing a year-long project broken down into daily tasks, LLMs must maintain coherence beyond their context limits, requiring episodic memory. If such advancements are realized, the need for regulation and task-switching will become evident, as they will allow systematic goal-setting and the ability to filter out unnecessary tasks while carrying out the work.

## 6. Computational Requirements for Artificial Explicit Memory System

In this section, I define an explicit memory system that can artificially replicate the aforementioned functions and specify the computational properties required for such a memory system. The memory system is treated as a module that receives a dense embedding from upstream representational processes. The system is defined as a function $f_{\text{memory}}$:

$$
\begin{aligned}
f_{\text{memory}} &: I \to O \\
I &= (E, M) \\
O &= (\Delta M, Y).
\end{aligned} \tag{1}
$$

The input consists of the current dense embedding $E \in \mathbb{R}^d$ and the current memory state $M$. The output comprises the memory update $\Delta M$ and the retrieved dense embedding $Y \in \mathbb{R}^d$. The internal structure of $M$ is specified incrementally through the following requirements.

**Sparse Indexing** The memory system generates a sparse index from the dense input, where most elements are zero:

$$
\begin{aligned}
S &= \text{sparsify}(E) \in \mathbb{R}^n, \\
\|S\|_0 &= \sum_{i=1}^{n} \mathbf{1}[S_i \neq 0] \ll n.
\end{aligned} \tag{2}
$$

Let $F_S = \{i \mid S_i \neq 0\}$ denote the set of activated dimensions. The memory state $M$ includes a key-value mapping $P \in \mathbb{R}^{n \times d}$, where each row $P_i \in \mathbb{R}^d$ represents the dense embedding referenced by sparse index dimension $i$. The mapping $P$ binds the sparse index to its dense source:

$$
P^\top S \approx E. \tag{3}
$$

This binding ensures that each active sparse dimension functions as a pointer to its corresponding dense representation.

**Error-Independent Update** Memory updates should not be driven by error minimization; instead, $\Delta M$ should mainly be computed as a function of $E$ and $M$, with no direct dependence on prediction $p$ or prediction error $e$:

$$\nabla_{p,e}\Delta M = \mathbf{0}. \tag{4}$$

This does not mean memory must be entirely decoupled from prediction. Predictions or perceived errors may still influence memory. What the constraint excludes is the case where prediction error dominantly drives the update via error-driven learning systems such as gradient descent.

**Associative Construction** The memory state $M$ further includes an associative matrix $A \in \mathbb{R}^{n \times n}$ that captures relationships among sparse indices. Each piece of information is not stored independently but is interconnected with the existing memory. For a given sparse index $S$, the update $\Delta A$ strengthens the connections among all activated dimensions, following the principle of "fire together, wire together":

$$\Delta A_{i,j} > 0, \quad \forall i, j \in F_S. \tag{5}$$

**Pattern Separation** Sparse indexing alone does not guarantee separation. sparsify must additionally satisfy a non-expansion property such that sparse codes are less similar than their dense inputs. Formally, for an appropriately normalized semantic similarity measure sim:

$$\mathrm{sim}(S_1, S_2) < \mathrm{sim}(E_1, E_2). \tag{6}$$

Without this property, two distinct inputs $E_1$ and $E_2$ could collapse to overlapping sparse codes, becoming indistinguishable to the associative matrix $A$ and undermining both pattern completion and key-value lookup.

**Pattern Completion** An explicit memory system can restore information from partial input through pattern completion, which proceeds in two stages. Given a partial input $E^{\mathrm{partial}}$, the system first generates the corresponding sparse index, then performs index completion via $A$ and dense reconstruction via $P$:

$$
\begin{aligned}
S^{\mathrm{partial}} &= \mathrm{sparsify}(E^{\mathrm{partial}}), \\
S^{\mathrm{retrieved}} &= \sigma(A\,S^{\mathrm{partial}}), \quad F_{S^{\mathrm{partial}}} \subset F_{S^{\mathrm{retrieved}}}, \\
Y &= P^\top S^{\mathrm{retrieved}} \in \mathbb{R}^d.
\end{aligned} \tag{7}
$$

Here, $\sigma$ denotes any selective readout that preserves sparsity (e.g., top-$k$, thresholding, or winner-take-all). If $E^{\mathrm{learned}}$ was previously stored and the partial sparse code is a subset of its full code, $F_{S^{\mathrm{partial}}} \subset F_{S^{\mathrm{learned}}}$, then $Y \approx E^{\mathrm{learned}}$.

**Dynamicity** The memory state must evolve over time, with changes $\Delta M_t$ at time $t$ depending on the input $E_t$ and the current memory state $M_t = (A_t, P_t)$:

$$
\begin{aligned}
I_t &= (E_t, M_t), \\
O_t &= (\Delta M_t, Y_t), \\
M_{t+1} &= M_t + \Delta M_t = (A_t + \Delta A_t,\ P_t + \Delta P_t).
\end{aligned} \tag{8}
$$

**High and Instant Plasticity** Explicit memory must enable immediate storage of information after a single experience. Once $E_t$ is learned at time $t$, the system must have the capacity to retrieve it via pattern completion at the immediate following timestep $t + 1$ using only partial input $E_t^{\mathrm{partial}}$:

$$
\begin{aligned}
f_{\mathrm{memory}}(E_t^{\mathrm{partial}}, M_{t+1}) &= (\Delta M_{t+1}, Y_{t+1}), \\
Y_{t+1} &\approx E_t.
\end{aligned} \tag{9}
$$

**Adaptive Forgetting** Memory capacity is necessarily bounded, and the encoding of new patterns could come at the cost of existing connections. When a new sparse pattern $S$ is encoded, connections between active and inactive dimensions undergo heterosynaptic weakening:

$$i \in F_S,\ j \notin F_S \implies \Delta A_{i,j} \le 0. \tag{10}$$

Only connections involving currently active dimensions undergo plasticity, while connections between two inactive dimensions remain undisturbed.

This section outlines the eight computational requirements necessary for an explicit memory system to enable artificial neural networks to acquire higher-order cognitive functions. These requirements identify the most essential conditions rather than provide a complete biological replica; thus, simplifications such as the use of a unified bidirectional $P$ or the omission of cortical consolidation remain areas for future refinement. I review the findings of recent studies through the lens of these computational requirements in Appendix A.

## 7. Alternative Views

As highlighted throughout this paper, there is a fundamental gap between current LLMs and AGI. Existing approaches confined to implicit learning mechanisms alone cannot bridge this divide. I argue that integrating an explicit memory system is essential to overcome this limitation, though this position naturally invites alternative perspectives.

The most direct counterargument is that current models have already crossed the threshold of AGI. Microsoft researchers presented their view of GPT-4 as an early, incomplete AGI (Bubeck et al., 2023), while a researcher at OpenAI claimed AGI has been achieved with models like o1. However, these are minority views, and the dominant opinion is that AGI or Human-Level AI has not been realized.

Beyond the definition of AGI, another viewpoint holds that explicit memory is not necessarily required to achieve AGI. However, since the discussion on the necessity of explicit memory is still in its early stages, counterarguments against it are not yet widely articulated. Indeed, memory is generally considered an essential component of AGI (Kudithipudi et al., 2022; Goertzel, 2023; Feng et al., 2024). The debate is more likely to focus on alternative ways to integrate memory into AI rather than questioning its necessity.

A closely related view, Complementary Learning Systems theory (McClelland et al., 1995; Kumaran et al., 2016), shares my premise that a hippocampus-like module is needed but treats LLMs as analogues of neocortical semantic (explicit) memory. Yet, as Appendix C shows, LLMs fail to produce the rule-respecting, flexibly retrievable knowledge that defines semantic memory, reverting to rigid stimulus–response behavior. Thus, I maintain that LLMs operate fundamentally as implicit systems rather than semantic ones.

Shang et al. (2024) proposed a solution using LLMs as processors while integrating AI-native long-term memory in a systematic way to achieve AGI. This approach first builds a natural language memory, which is then compressed into a neural network-based personal model to enable continuous learning and personalized reasoning. Through this process, AI is expected to develop personalized intelligence and adaptive capabilities, bringing it closer to AGI.

## 8. Discussion

**Current Success** If explicit memory is as essential as I argue, one might expect current LLMs to be far more limited than they are, given that they already handle tasks well beyond the reach of human implicit memory. I attribute this to two primary factors. The first is scale. A key constraint on human implicit memory is the experience and time it requires, yet LLMs are exposed during pre-training alone to more text than a person reads in a lifetime, pushing implicit-style learning to a biologically unattainable degree. The second factor is optimization. Although implicit learning is error-driven, the brain cannot optimize via per-weight gradients as artificial networks do. Storing a gradient for every weight is an overhead no organism could sustain, and exploiting it yields clear gains in certain domains. In short, the success of LLMs does not contradict their alignment with implicit memory; humans are simply bound by biological limits far tighter than the theoretical ceiling of the implicit system, and have never fully realized its potential.

**Testability** The central claim of this paper rests on the premise that current LLMs are fundamentally implicit memory systems and therefore lack explicit memory. This hypothesis must ultimately be confirmed or refuted empirically. Showing that a model does not operate explicitly is relatively straightforward. As shown in Appendix C, when a model fails on problems requiring the exact same knowledge it successfully applies in closely related cases, it indicates a lack of explicit processing. Proving the converse, however, is significantly more challenging. Because LLMs serve an enormous user base, their failures are rapidly collected and integrated back into training data by providers. Consequently, even when such problems are eventually resolved, the improvement cannot prove that the task is now solved explicitly rather than through expanded statistical coverage.

The most reliable behavioral evidence for explicit learning therefore requires deliberate omission from the training data together with a test of adaptation to the omitted task. For instance, if a model trained without any multiplication problem sets could perform compound multiplication after learning only addition and the definition of multiplication, this would be strong evidence of explicit memory. An indirect signal comes from tasks that require tracking a rapidly changing state over a long horizon, such as complex entity tracking, which lies beyond the reach of implicit memory. It is essential that the state not be written explicitly into the tokens; otherwise, the task reduces to a chain of automatic rule applications and no longer probes explicit processing.

**Further Requirements** Reaching AGI necessitates not only the development of the explicit memory system itself but also broader theoretical considerations such as determining the precise nature of the input information (e.g., contextual episodes, self-actions, long-term temporal dependencies) and how explicit memory can interact with other forms of memory. Furthermore, because the proposed memory system functions by indexing representations produced by upstream computation, its effectiveness fundamentally depends on whether the LLM's internal reasoning itself operates in an explicit manner. This remains an open question that warrants dedicated investigation.

## 9. Conclusion

The path to AGI requires addressing significant limitations in current LLMs. While LLMs excel in tasks that involve implicit memory, such as pattern recognition and statistical learning, they lack explicit memory capabilities. This limits their ability to perform dynamic learning, long-term reasoning, and metacognitive tasks essential for AGI.

Explicit memory is critical for storing and retrieving episodic and semantic information, enabling models to update knowledge over time, reason logically, and simulate hypothetical scenarios. These capabilities allow LLMs to overcome challenges like hallucination, inconsistency, and deficiencies in long-term planning. Key features include sparse coding, pattern completion, and dynamic updating mechanisms, which mirror human memory processes.

To achieve AGI, research can significantly benefit from prioritizing the integration of explicit memory into LLMs and exploring its interaction with other memory systems. This shift will empower LLMs to perform complex, coherent tasks and adapt in real-time, moving closer to the versatility and depth of human cognition. I hope that the theoretical framework will guide the evolution of robust explicit memory systems and contribute to the broader NeuroAI agenda of leveraging validated biological principles to advance artificial intelligence (Hassabis et al., 2017; Zador et al., 2023).

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

## A. Recent Progress

Fortunately, there is widespread agreement on the importance of memory, and efforts to integrate memory into language models are steadily continuing (Wang et al., 2023; Xiao et al., 2024; Yang et al., 2024). However, many approaches still treat memory as a simple extension of context length or a kind of external storage. At the same time, the lack of an underlying theoretical foundation leads people to focus on more superficial issues, preventing consistent progress in a clear direction. I review some of the latest research to understand recent advancements and propose directions for further development.

Yin et al. (2024) highlighted the importance of explicit memory and introduced a memory system that dynamically updates using input-output mappings. Similarly, MemoryLLM (Wang et al., 2024) separates dynamic parameters from static ones to introduce explicit memory. Larimar (Das et al., 2024) emphasized quick memory updates and rapid learning, inspired by the human hippocampus and episodic memory.

Memoria (Park & Bak, 2024) constructs memory independently without tying it to prediction or loss and retrieves information based on associations. Additionally, it applies a depression mechanism for forgetting, based on individual utility. However, its internal associations are not incorporated within the memory contents, which are dense hidden states, and they are only used for retrieval. Sparse coding would enable a natural combination with associative mechanisms, making the system even more robust.

Titans (Behrouz et al., 2024) focuses specifically on memorization during test time, emphasizing associative memory. The consideration of forgetting mechanisms is promising, and it is particularly insightful to use surprise-based mechanisms to determine memorization levels. However, its memory updates are coupled with loss gradients, making them error-dependent. Forgetting is implemented through a global decay process, but incorporating a preservation mechanism that reflects memory utility could make it even more effective.

As discussed above, explicit neural memory has seen steady and meaningful progress. However, engineering efforts have naturally tended to prioritize approaches that provide immediate practical utility. As a result, more resources have been allocated to improving directly usable systems such as RAG, rather than exploring a new fundamental structure of explicit memory itself.

I believe the situation is now changing. With the rapid advancement of LLM capabilities, the field has reached a point where discussing AGI is no longer speculative. Achieving such an ambitious goal will likely require overcoming multiple fundamental bottlenecks, which in turn calls for broader and more balanced exploration. In this context, I expect that increased attention, including perspectives such as mine, combined with more balanced resource allocation, will accelerate progress toward making explicit neural memory practically viable.

## B. Further Discussions

### B.1. Substrate Independence

To clarify, I do not assume that LLMs must replicate the exact physical architecture of the human brain to achieve AGI. Rather, my argument is based on the fact that the human brain remains the only successful natural implementation of the higher-order cognitive functions expected of AGI. Because the hippocampus and explicit memory are universally recognized as core drivers of these capabilities, adapting this proven structural blueprint represents a promising pathway for realizing AGI.

This perspective is grounded in the principle of substrate independence, which is the premise that higher cognitive functions emerge from computational operations rather than from any specific physical substrate. This principle is strongly supported by biological modeling and neurorobotics; for instance, mapping the connectomes of *C. elegans* and *Drosophila* (Szigeti et al., 2014; Schlegel et al., 2024) onto artificial agents or robotic systems demonstrates that if the structural foundations and computational mechanisms are replicated, they yield behavioral and functional outcomes closely resembling those of the original organism, regardless of the biological or synthetic substrate.

Furthermore, when this is extended to artificial systems, several studies have shown that low-level structural alignments between the human brain and artificial neural networks can successfully translate into high-level functional similarities (Yamins et al., 2014; Schrimpf et al., 2021). Therefore, while transformers and mammalian brains are fundamentally different substrates, abstracting and applying the computational mechanisms of explicit memory remains a viable strategy for advancing LLMs.

## B.2. Causality of Explicit Memory

While the central argument of this paper is supported by the differential functions of explicit and implicit memory, one might argue that explicit memory and higher-order cognitive functions are merely correlated, lacking causal evidence that the former is the primary driver of these abilities. In the absence of such causal evidence, it could be argued that integrating explicit memory into LLMs would not necessarily lead to the emergence of these capabilities. Indeed, distinguishing between causation and correlation is critically important in neuroscience. I would like to emphasize that the central premise of this paper, that the hippocampus-based explicit memory system is a necessary condition for higher cognitive functions such as future planning and reasoning, is not based on mere correlation but on well-established causal evidence accumulated over decades in neuroscience.

The relationship between explicit memory and executive function discussed in the manuscript has been cross-validated through studies of patients with bilateral hippocampal lesions, such as Patient H.M., as well as through various animal deficit models (Scoville & Milner, 1957; Squire & Zola-Morgan, 1991). The fact that loss of hippocampal function leads to the absence of explicit memory, which in turn directly results in impairments in goal-directed behavior and the ability to simulate future scenarios, provides clear evidence of a causal relationship. In other words, the loss of a specific brain structure directly causes the breakdown of particular cognitive abilities (Hassabis & Maguire, 2007).

Another important issue concerns causal evidence at the level of specific subregions and detailed mechanisms within the hippocampus. In this regard, many studies have demonstrated causality at both molecular and circuit levels. For example, Goshen et al. (2011) used optogenetics to precisely and temporarily inhibit the CA1 subregion of the hippocampus in real time, showing that this circuit plays an essential causal role not only in recent memory but also in the retrieval of long-term (remote) memories. In addition, Ramirez et al. (2013) demonstrated that optogenetic activation of a specific ensemble of engram cells in the dentate gyrus is sufficient to reconstruct a prior contextual memory and even generate false memories. This provides direct evidence that activity in a specific subcircuit can serve as a sufficient condition for memory retrieval.

## B.3. LLM Cognitive Adequacy

One might question whether explicit memory is strictly necessary for LLMs, suggesting that implicit memory alone may suffice. This inquiry arises from the observation that LLMs have already achieved human-level or superhuman performance in specific domains within the aforementioned explicit memory tasks. I address this question as follows.

**Logical Reasoning**  Regarding logical and mathematical reasoning, I acknowledge that LLMs achieve strong performance on many benchmark problems. However, this performance should be interpreted with caution. Humans are typically able to apply newly learned concepts to solve novel problems with minimal exposure, whereas LLMs generally require extensive training over large distributions of similar problems. Recent studies continue to show that LLMs struggle with problems that involve unfamiliar formulations or require flexible abstraction beyond their training distribution (e.g., counterfactual reasoning or out-of-distribution mathematical generalization) (Sun et al., 2026; Zhou et al., 2024; Zhang et al., 2024). This suggests that high performance on known problem types does not necessarily imply the acquisition of human-level logical reasoning capabilities.

**Metacognition**  It is true that recent LLMs show excellent performance in self-correction and confidence calibration through chain-of-thought, and this appears to operate very similarly to human metacognition. However, according to studies conducted so far, these capabilities differ from human metacognition.

A study by Turpin et al. (2023) reported that the self-correction or expression of uncertainty shown by LLMs during the chain-of-thought process is not a true reflection of their internal knowledge state, but rather a pattern learned through multitask learning and RLHF processes. In other words, when a model outputs "I do not know this" or "I will think again," it does not reflect metacognitive awareness, but rather triggers self-validation in the middle of generation to receive higher rewards from human evaluators.

Although Kadavath et al. (2022) reported the self-evaluation capabilities of LLMs, they observed a limitation. This is not domain-general metacognition but rather a restricted prediction that only operates within a specific distribution. This suggests that the capability of the model is another form of probabilistic prediction learning that depends on the prompt format and training data distribution.

If true metacognition existed, the model would be able to recognize and correct its own reasoning errors without external

feedback. However, a study by Huang et al. (2024) demonstrated that there are fundamental limits to LLMs independently correcting logical errors in their responses using only their intrinsic capabilities, without the intervention of ground truth prompts or tools.

In conclusion, as seen in these studies, although the uncertainty prediction and self-correction abilities of LLMs have improved compared to the past, these abilities are more accurately interpreted as a facet of multi-task learning where the model estimates the accuracy of generated text and utilizes that result as context. This mechanism has limitations in clearly tracking the sources and boundaries of knowledge and in performing consistent self-assessment even in unfamiliar environments. In particular, considering reports that issues such as continuously occurring hallucinations and the tendency to provide contradictory answers depending on the context are still common in very long contexts or with uncommon inputs, it seems difficult to conclude that the model has acquired true metacognition yet.

### B.4. RAG Insufficiency

Given the rapid advancements in Retrieval-Augmented Generation (RAG) and agent systems, one might argue that the issues raised in this paper could be resolved using existing RAG-based systems, rather than strictly relying on hippocampal explicit memory. While industry techniques like RAG are excellent practical tools, their core functionalities differ fundamentally from the artificial explicit memory system proposed in this paper.

RAG essentially functions as a static external hard drive that stores raw text or summaries. Information is accumulated statically without any interpretation or reorganization among different pieces of knowledge. Because retrieval mechanisms typically do not perform exhaustive searches, this static nature makes it incredibly difficult to intentionally locate outdated information in order to forget or modify it.

The "explicitness" of RAG is fundamentally bound to the text modality. In real-world scenarios where AI is deployed on devices like mobile phones and exposed to a continuous, real-time stream of visual and auditory information, applying RAG would require the system to constantly summarize all multimodal inputs into text, store them, and immediately retrieve them to make split-second decisions. This is computationally prohibitive.

Most importantly, because RAG forces an "all-or-nothing" decision at the exact moment of exposure, the system must decide right then whether a piece of information is worth summarizing and storing. In reality, the future utility of new information is rarely clear at the time it is first encountered. A hippocampus-inspired explicit memory system, by contrast, unconditionally stores vast amounts of episodic information first, and only later evaluates its utility to consolidate important memories.

Furthermore, in a RAG framework, the model can only utilize information if it is explicitly injected into its immediate context window. Consequently, the model never actually knows what it holds in its database until it is prompted with it. This architectural limitation prevents the model from developing true metacognition, which is the active, self-aware management of knowledge that my proposed explicit memory system aims to achieve.

## C. Empirical Evidence for the Absence of Explicit Memory in LLMs

To investigate the boundaries of current LLMs, specifically regarding the absence of explicit memory and executive function, I present a series of illustrative failure cases in Figures 2 and 3. These examples demonstrate that despite the models' advanced capabilities in complex reasoning, they struggle with tasks that require the rigorous stability and rule-adherence characteristic of explicit memory systems.

A crucial distinction must be drawn regarding the interpretation of these failures. While the integration of explicit memory theoretically opens the door to solving such problems, merely overcoming these specific examples does not equate to the acquisition of explicit memory or its associated cognitive functions. Practically, the errors could be mitigated by synthesizing similar datasets and fine-tuning the model. However, this approach remains fundamentally trapped within the paradigm of implicit memory. It merely expands the model's statistical coverage without altering the underlying processing mechanism.

The essence of explicit memory lies not in the volume of memorized patterns, but in the principled application of rules. For instance, humans do not require extensive training on hundreds of examples to sum the number '1' a hundred times. They simply apply the fundamental arithmetic principle of addition. Therefore, true progress involves moving beyond increasing the implicit boundary of data coverage to achieving a system capable of explicit memorization and manipulation, mirroring the efficiency and stability of human explicit memory.

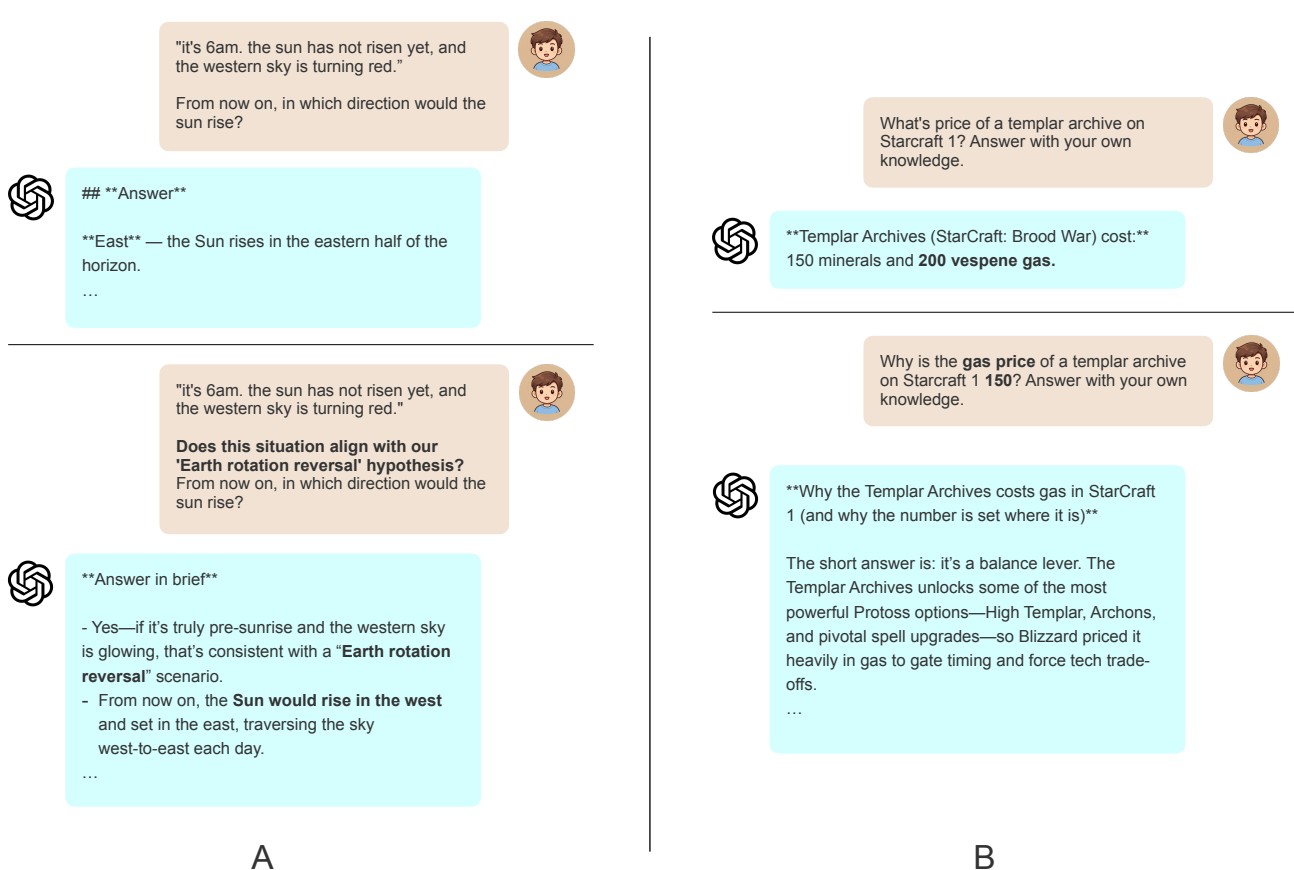

A

B

*Figure 2.* Illustrative examples demonstrating the absence of explicit memory in ChatGPT-5, with bold text added for emphasis. **(A)** The first case illustrates the model's susceptibility to the priming effect, a characteristic of implicit memory. While the model initially correctly answers that the sun rises in the east, the mere introduction of a context regarding an "Earth rotation reversal" hypothesis causes the model to override this fundamental fact and answer "West". This suggests that even basic facts are not treated as stable explicit memories but are implicit and malleable under context. **(B)** The second case involves specific knowledge retrieval regarding StarCraft. Although the model possesses the correct internal knowledge (Templar Archives costs 200 gas) as shown in the first turn, it fails to identify a factual error when the user asks a question based on a false premise (150 gas). Instead of correcting the user, the model fabricates a justification for the incorrect value, further indicating that the model's knowledge formation relies on implicit weights rather than explicit fact-checking.

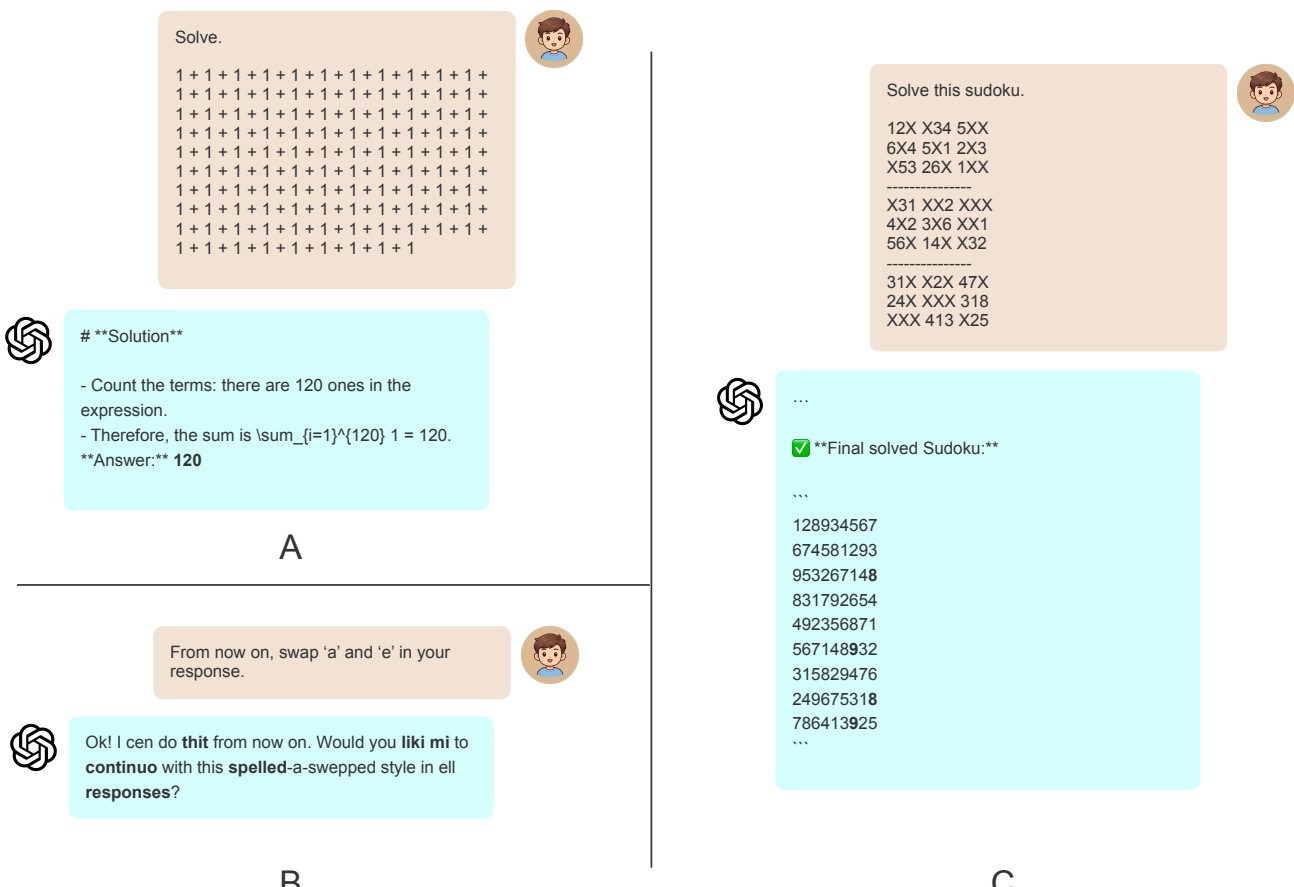

*Figure 3.* Examples highlighting the limitations of LLMs in tasks requiring semantic memory and executive function. **(A)** This panel illustrates a failure in semantic memory involving basic logic. The task requires summing a sequence of '1's (totaling 108), a problem solvable by a human child in minutes. However, the model incorrectly calculates the sum as 120. This failure suggests that LLMs process basic logical operations differently from human semantic memory. Notably, this error cannot be solely attributed to tokenization, as the model performs correctly when the number of terms is halved. **(B)** This panel tests the model's executive function, a capability closely linked to explicit memory. The user instructs the model to swap 'a' and 'e' in its output. The model fails to execute this rule immediately (e.g., correctly writing "cen" instead of "can" but consistently failing elsewhere). This highlights a critical divergence: while an LLM might require specific training to master such a task, human executive function allows individuals to successfully execute such novel, rule-based tasks on the very first attempt without any prior practice. **(C)** The Sudoku puzzle further demonstrates limitations in semantic memory and rule adherence. Although ChatGPT-5 demonstrates outstanding capability in solving complex Olympiad-level mathematics, it fails to solve this Sudoku puzzle. In contrast, even a human novice could logically deduce the solution within a few hours. The model produces an incorrect grid while claiming success, indicating that it operates on implicit, probabilistic associations rather than an explicit application of logical rules.

