# OpenReview forum: "Position: Hippocampal Explicit Memory Is the Cornerstone for AGI"
_ICML.cc/2026/Position_Paper_Track — ICML 2026 Position Paper Track regular_

### Official Review · Reviewer_Wovx · 2026-02-27

**Significance:** 3
**Argument Clarity:** 4
**Rating:** 5
**Confidence:** 4

**Questions:**

One thing that I find missing in this position paper is some connection to existing architectures that could potentially facilitate the creation of an explicit memory module. One key component that the authors highlight is sparse coding. Do the authors believe that existing "sparse" architectures such as Sparse Autoencoders, Winner-Takes-All (WTA/LWTA) or Spiking NN rationales are sufficient for implementing the proposed explicit memory module? Alternatively, do the authors posit that we need a new paradigm that is disentangled from the conventional architectural and training regimes to achieve such a behavior?

Integrating a high-plasticity, one-shot learning module for explicit memory with a slow-learning dense model such as an LLM could create stability issues. How do the authors propose managing the potential of catastrophic interference or unpredictable behavior during the memory update process?

Regarding the update of the memory state the authors initially focus on heavily reinforcing the associations between the neurons, in a "fire-together, wire together" rationale; Could the authors provide a discussion on the specific triggers for Long-Term Depression in their  approach? Specifically, when should the system prioritize "forgetting" or weakening an association (the "use it or lose it" principle)? While the authors mention this occurs when neurons do not fire together over a "sufficiently long time interval," how might this interval be computationally determined or parameterized?

One final issue that needs to be addressed in the novelty and differentiation of this work. How does this position paper significantly differs from recent cited works, e.g., Shang et al (2024), Wang et al. (2025) or Song et al (2025), which also explore the integration of episodic/semantic memory into AI? The alternative views section provides only a limited discussion on these papers; however, they appear to share several fundamental similarities with this work. A more detailed comparative analysis in this context is imperative to clarify the paper's unique contribution.

**Alternative Views Section:**

Yes

**Compliance With Llm Reviewing Policy A Conservative:**

Affirmed.

**Discussion Potential:**

3

**Final Justification:**

In my view, most of the core issues of the reviewers have been addressed by the authors. Given that the suggested improvements will be incorporated in the final manuscript, I retain my score to 5.

**Paper Summary:**

This position paper argues that the current limitations of Large Language Models, i.e., hallucinations, logical reasoning issues and planning inconsistencies, stem from the lack of explicit memory or equivalently their exclusive reliance on implicit memory mechanisms. In this context, the authors posit that to achieve Human Level AI (HLAI) and/or AGI, the integration of an explicit memory system is necessary.

The authors draw from extensive neuroscience research to highlight the differences between explicit and implicit memory systems in the mammalian brain, noting their respective contributions in processes such as logical reasoning, executive function and meta-cognition. The paper also provides a detailed discussion on why the current LLM architectures are more aligned with implicit memory.

Based on this analysis, a computational framework for an explicit memory system is proposed to replicate the essential functions and adress the challenges that current LLMs face, while briefly acknowledging alternative views in the field.

**Position:**

Yes

**Position In Title:**

Yes

**Related Work:**

3

**Strengths And Weaknesses:**

The paper successfully bridges neuroscience works and AI, drawing on established biological principles such as sparse coding, plasticity, and long term depression, to address current LLM bottlenecks. The authors provide a formal computational requirement list for an explicit memory module that appears to be feasible, though it contains some missing links (detailed below).

The paper is overall well-written and easy to follow. The transition from the biological background to the proposed position is well-handled and smooth, rendering the treated subject very accessible while also remaining thorough. The authors also acknowledge alternative views, e.g., that AGI is already achieved or that hybrid systems may not require biological fidelity.

One key hypothesis of the paper is the assumption that because the human brain requires/uses explicit memory, an artificial system like LLMs, must follow the same architectural blueprint for HLAI. However, LLMs and the mammalian bran are fundamentally different substrates; forcing an explicit memory structure onto a transformer like architecture might result in unpredictable or sub-optimal behaviors.

In this context, while the "what" and the "why" are clearly defined in the main text, the "how" remains somewhat abstract. The authors propose the seven computational requirements for explicit memory but leave the interaction between these modules and existing architectures largely unexplored. At the same time, the argument heavily relies on the assumption that the brain's memory mechanisms are sufficiently understood, despite ongoing debates about the mechanisms of the memory system.

Finally, even though some alternative views are provided, I find the discussion to be limited, especially regarding methods that closely align to what the position paper argues.

**Support:**

3

---

> ### Author Rebuttal · Authors · 2026-03-25
>
> We would like to appreciate the reviewer for taking the time to review our research.
>
> **W1.** To clarify, our argument is based on the fact that the human brain is the only successful implementation of the higher-order cognitive functions expected of AGI. Because the hippocampus is a core of these, adapting this proven structure represents a promising pathway.
>
> This is grounded in the principle of substrate independence. For instance, research on *C. elegans* and *Drosophila* [1, 2] suggests that if the computational mechanisms are replicated, they yield identical behaviors. Furthermore, several studies have shown that low-level structural alignments between the brain and AI can translate into functional similarities [3, 4]. While transformers and brains are fundamentally different, applying the computational mechanisms of explicit memory remains a viable strategy for advancing LLMs.
>
> [1] Szigeti, B., et al. (2014). OpenWorm: an open-science approach to modeling Caenorhabditis elegans. *Frontiers in computational neuroscience*.
>
> [2] Schlegel, P., et al. (2024). Whole-brain annotation and multi-connectome cell typing of Drosophila. *Nature*.
>
> [3] Yamins, D. L. K., et al. (2014). Performance-optimized hierarchical models predict neural responses in higher visual cortex. *Proceedings of the National Academy of Sciences*.
>
> [4] Schrimpf, M., et al. (2021). The neural architecture of language: Integrative modeling converges on predictive processing. *Proceedings of the National Academy of Sciences*.
>
> ---
>
> **W2.** We aim to introduce a novel, well-grounded perspective by bridging neuroscience and AI literature. We have elaborated on this scope further in our response to Reviewer YehM.
>
> Regarding interaction, since the hippocampal-striatal circuit is fundamental, we will expand upon this to add a discussion.
>
> Regarding the completeness of brain memory systems, we acknowledge that it is not perfectly understood, but neuroscience has made profound advancements through optogenetics and large-scale neural recording. Core mechanisms like hippocampus-neocortex interactions are now backed by substantial experimental evidence. We believe that introducing these established principles to the ML community holds immense innovative potential.
>
> ---
>
> **W3.** We address methodological studies closely aligned with our claims in Appendix A.
>
> ---
>
> **Q1.** We consider Sparse Autoencoders (SAEs) to be promising. SAEs conceptually align well with memory indexing theory. Vivid experiences travel from the entorhinal cortex to the dentate gyrus (DG). The DG employs strong inhibitory mechanisms so that only a tiny fraction of granule cells become excited. Given this functional parallel, the DG can be understood as analogous to the sparse embedding of an SAE.
>
> However, SAEs are trained to compress a dense embedding space whose semantic structure is already known. Conversely, DG neurons do not have predetermined semantic roles. In reality, an agent cannot have prior knowledge of its future experiences. Therefore, we have to use a new paradigm together.
>
> ---
>
> **Q2.** We believe interference issues can be resolved structurally. In the human brain, stability is managed through spatial and temporal separation. Fast-learning explicit memory and slow-learning implicit memory are stored in distinct regions. Because of this speed difference, the implicit system remains highly stable even as explicit memory undergoes rapid modifications.
>
> ---
>
> **Q3.** Based on the principles of Spike-Timing-Dependent Plasticity (STDP), synaptic strengthening and weakening are triggered by the timing of neuron activations. Conceptually mapping this to our model, the trigger for the forgetting is the activation of a neuron itself. When a neuron activates, it can weaken the connections from the neurons that haven't activated for the previous $w$ timesteps. Regarding its scale, the effective time window for STDP is typically on the order of tens of milliseconds. Using a small scalar is sound.
>
> ---
> **Q4.** While they are valuable works on AI memory, our core contributions differ significantly. Wang and Shang share our view that simple context expansion or RAG is insufficient for AGI. However, they approach this primarily through engineering and architectural solutions, such as Wang's four-module cognitive system or Shang's LoRA-based dynamic memory. In contrast, our paper analyzes a fundamental mechanism specifically by comparing it with humans'.
>
> Song et al. broadly categorize LLM reasoning failures across various factors, such as training data or tokenization. Our research, however, is sharply focused exclusively on memory-related cognitive failures, providing a much deeper analysis.
>
> In short, the main difference is that they viewed LLMs from an AI perspective, whereas we compared them directly to humans. Our unique contribution lies in using a mechanistic comparison between LLMs and human memory to justify the necessity of hippocampal explicit memory for realizing AGI.

---

> > ### Author Rebuttal · Reviewer_Wovx · 2026-04-01
> >
> > I would like to thank the authors for their comprehensive responses. After reviewing the feedback provided to all reviewers, I believe the authors have effectively addressed the core issues raised. I agree with the points regarding precise language and structural improvements suggested by the other reviewers (e.g., Rev. cNoq), as well as incorporate further discussions on related works and the differentiation of this work; these will significantly strengthen the manuscript. I encourage the authors to integrate these insights into the final version. Given these improvements, I have increased my score.

---

### Official Review · Reviewer_tM43 · 2026-03-12

**Significance:** 3
**Argument Clarity:** 3
**Rating:** 4
**Confidence:** 4

**Questions:**

1. What are the essential differences between the “artificial explicit memory system” defined in the paper and existing industry solutions (such as RAG)?

2. The authors suggest that hallucinations stem from a lack of episodic metadata. Even with explicit memories, if the memory itself stores erroneous information, how can the system self-correct through metacognition?

3. Given that the authors have not referenced the work mentioned in the Weaknesses, could the authors please explain the differences between this paper and that work?

**Alternative Views Section:**

Yes

**Compliance With Llm Reviewing Policy A Conservative:**

Affirmed.

**Discussion Potential:**

3

**Final Justification:**

My concerns has been addressed.

**Paper Summary:**

This position paper argues that the current trajectory of artificial intelligence, which is dominated by Large Language Models (LLMs), is limited because it relies almost exclusively on implicit memory mechanisms—analogous to the basal ganglia in the human brain. The authors contend that achieving Human-Level AI (HLAI) requires integrating explicit memory, modeled on the hippocampal system. To substantiate this argument, they conduct a detailed comparative analysis of the characteristics of explicit versus implicit memory, contrasting them with the features of LLMs. Furthermore, the authors conclude by defining a set of computational requirements for artificial explicit memory systems to guide future architectural innovations.

**Position:**

Yes

**Position In Title:**

Yes

**Related Work:**

2

**Strengths And Weaknesses:**

Strengths:

- The topic of the paper is highly relevant and significant to the current LLM boom, successfully bridging neurobiological theories of memory with contemporary AI challenges. It provides a framework for understanding why LLMs struggle with long-term consistency and factual grounding.

- The argument is logically sound, clearly delineating the differences between explicit and implicit memory in terms of their formulation mechanisms, retrieval processes, and cognitive functions. It also highlights the similarities between the learning process of LLMs and implicit memory.

- The paper offers an interesting interpretation of the hallucination problem in LLMs, suggesting that its root cause lies in the absence of “episodic metadata about knowledge”—that is, a lack of explicit memory support. Moving beyond argument, the paper also attempts to define the computational requirements for artificial explicit memory systems, providing a theoretical framework for future development.

Weaknesses:

- As a position paper, it relies heavily on logical deduction and literature. Although some examples are provided in the appendix, there is a lack of quantitative evaluation or comparative experiments, even on a small scale, to support their arguments.

- The industry has seen numerous attempts to develop explicit memory systems, such as retrieval-augmented generation (RAG) and context management. While the authors acknowledge related progress in the Appendix, we think they have not sufficiently clarified how these differ from the "artificial explicit memory system" they propose.

- The views presented in the paper are not entirely novel. Another review work provides a more comprehensive discussion [1].

[1] Zixia Jia, et al. [The AI Hippocampus: How Far are We From Human Memory?](https://openreview.net/forum?id=Sk7pwmLuAY) TMLR 2025.

**Support:**

2

---

> ### Author Rebuttal · Authors · 2026-03-30
>
> We would like to appreciate the reviewer for taking the time to review our research.
>
> **W1.** As a position paper, our primary goal is to introduce a novel perspective by synthesizing neuroscience and AI literature, providing a conceptual roadmap for future research rather than presenting a specific empirical methodology. We have provided a detailed discussion regarding this focus and our guiding principles in our response to Reviewer YehM.
>
> ---
>
> **W2, Q1.** While industry techniques like RAG are excellent practical tools, their core functionalities differ fundamentally from the artificial explicit memory system proposed in our paper.
>
> RAG essentially functions as a static external storage that stores raw text or summaries. Information is accumulated statically without any interpretation or reorganization among different pieces of knowledge. Because retrieval mechanisms typically do not perform exhaustive searches, this static nature makes it incredibly difficult to intentionally find out and forget or modify outdated information.
>
> The "explicitness" of RAG is fundamentally bound to the text modality. In real-world scenarios where AI is deployed on devices like mobile phones and exposed to a continuous, real-time stream of visual and auditory information, applying RAG would require the system to constantly summarize all multimodal inputs into text, store them, and immediately retrieve them to make split-second decisions. This is computationally prohibitive.
>
> Most importantly, because RAG forces an "all-or-nothing" decision at the exact moment of exposure, the system must decide right then whether a piece of information is worth summarizing and storing. In reality, the future utility of new information is rarely clear at the time it is first encountered. A hippocampus-inspired explicit memory system, by contrast, stores vast amounts of episodic information first, and only later evaluates its utility to consolidate important memories.
>
> Furthermore, in a RAG framework, the model can only utilize information if it is explicitly injected into its context window input. Consequently, the model never actually knows what it holds in its database until it is prompted with it. This architectural limitation prevents the model from developing true metacognition, which is the active, self-aware management of knowledge that our proposed explicit memory system aims to achieve.
>
> ---
>
> **W3, Q3.** The referenced work is an excellent and comprehensive review that masterfully organizes the vast amount of research on LLM memory systems. However, our paper and the referenced work diverge entirely in terms of defining scope, cognitive science perspective, and core contribution. The referenced paper defines memory in a broad sense, encompassing all the ways information is stored and utilized in LLMs, and it focuses on organizing the widest possible variety of engineering and empirical studies.
>
> The most significant difference is that the referenced paper does not adopt a perspective that compares LLMs with human memory systems. It appears that this paper compared LLM with human memory due to the title and the use of terms like explicit memory, implicit memory, and the hippocampus. Yet, the main text briefly uses these terms in the introduction to explain abstract concepts, and it does not maintain a comparative perspective between artificial networks and human memory.
>
> Looking at their classification, they treat topics like text, graphs, and RAG as explicit memory, while treating transformer architectures or scaling laws as implicit memory. Furthermore, concepts like Agentic memory and MLLM memory are introduced on the same level as implicit and explicit memory. This indicates that these terms are used for the authors' categorization of existing methodologies rather than as strict neuroscientific definitions. Ultimately, the memory referred to in that paper is closer to a broad concept within AI and has no direct correlation with human memory.
>
> In contrast, our paper directly applies the strictly defined memory systems from neuroscience to the underlying operating mechanisms of LLMs and conducts a mechanistic comparative analysis. Through this, we establish an academic framework that interprets artificial memory by connecting it with human memory.
>
> ---
>
> **Q2.** Metacognition enables self-correction by identifying contradictions between stored knowledge and new external information. When such conflicts arise, the system utilizes episodic metadata to evaluate the relative credibility of both sources. This allows the model to assess the reliability of its internal data against new evidence and recalibrate or correct erroneous memories accordingly.

---

> > ### Author Rebuttal · Reviewer_tM43 · 2026-04-07
> >
> > Thank you for the responses. They fully addressed my concerns.

---

### Official Review · Reviewer_YehM · 2026-03-13

**Significance:** 4
**Argument Clarity:** 4
**Rating:** 5
**Confidence:** 4

**Questions:**

- Why hasn't explicit neural memory succeeded thus far? How can we make explicit neural memory work?

**Alternative Views Section:**

Yes

**Compliance With Llm Reviewing Policy A Conservative:**

Affirmed.

**Discussion Potential:**

4

**Final Justification:**

The paper is well written and the arguments make sense logically. I found the overall story to be compelling, and maintain that I believe the paper should be accepted.

**Paper Summary:**

The paper takes the position that explicit neural memory (as opposed to implicit neural memory of today's most powerful models) is necessary for human-level AI. They argue this by drawing a comparison to humans, and how the hippocampus in humans is a necessity for many tasks. The authors describe many different analogies between current models and the human brain (primarily the basal ganglia), and describe the flaws of current models using this analogy.

**Position:**

Yes

**Position In Title:**

Yes

**Related Work:**

4

**Strengths And Weaknesses:**

#### Strengths
- The argument was well presented, had strong evidence, followed a clear logical flow, had great reasoning, and was very convincing overall.
- The position taken by the authors is of importance for the greater community, as current models have little to no explicit memory, and it remains highly questioned whether current paradigms are sufficient for human-level intelligence.
- The position paper does a great review of the neuro/AI literature and makes clear and insightful analogies regarding the parallels of current models to structures within the human brain
    - The parallels drawn between the human brain and current models match the current laggings of models, and hence reinforce the strength of the argument.



#### Weaknesses
- The authors don't describe a way to achieve an explicit memory using current approaches, or why current explicit memory attempts have failed

**Support:**

3

---

> ### Author Rebuttal · Authors · 2026-03-30
>
> We would like to appreciate the reviewer for taking the time to review our research.
>
> **W1, Q1.** As you pointed out in your strengths, this work is a position paper whose primary goal is to argue that hippocampal explicit memory is a promising component for achieving HLAI. Accordingly, our main focus has been on providing a mechanistic perspective: we align the core learning dynamics of LLMs with the human implicit memory system and argue that their key limitations stem from the absence of an explicit memory system. Given this objective, we prioritized conceptual clarity and theoretical grounding over proposing a specifically defined method or implementation. Instead, in Section 6, we outlined high-level computational requirements as guiding principles, as we believe this broader perspective is more effective in encouraging diverse future explorations rather than constraining the space with a single concrete design. That said, we fully agree that progressing toward concrete methodologies is crucial, and we plan to validate our claims with empirical results in future work.
>
> Regarding current approaches, Appendix A discusses how existing methods attempt to approximate or utilize explicit memory from the perspective of this paper, including both their contributions and limitations. As noted there, explicit neural memory has not necessarily failed, but rather has seen steady and meaningful progress. However, engineering efforts have naturally tended to prioritize approaches that provide immediate practical utility. As a result, more resources have been allocated to improving directly usable systems such as RAG, rather than exploring the new fundamental structure of explicit memory itself.
>
> We believe the situation is now changing. With the rapid advancement of LLM capabilities, the field has reached a point where discussing AGI is no longer speculative. Achieving such an ambitious goal will likely require overcoming multiple fundamental bottlenecks, which in turn calls for broader and more balanced exploration. In this context, we expect that increased attention, including perspectives such as ours, combined with more balanced resource allocation, will accelerate progress toward making explicit neural memory practically viable.

---

> > ### Author Rebuttal · Reviewer_YehM · 2026-04-02
> >
> > My concerns have been addressed and I am keeping my score as a 5 for acceptance. After reading the other rebuttals, I still feel confident in the merit of the argument as well as the position being argued for in the paper.

---

### Official Review · Reviewer_aFMf · 2026-03-16

**Significance:** 2
**Argument Clarity:** 3
**Rating:** 4
**Confidence:** 4

**Questions:**

N/A

**Alternative Views Section:**

Yes

**Compliance With Llm Reviewing Policy A Conservative:**

Affirmed.

**Discussion Potential:**

2

**Final Justification:**

After reading the rebuttals, I remain unconvinced by their argument that the memory described in this paper is necessary for future foundation models. However, it seems that the authors are capable of addressing and rewriting my concerns about presentation and story cohesion.

With the assumption that the authors address my presentation concerns, I bump my score to a 4. Perhaps another reader will find the argument more convincing, and I do believe that the content of the paper is a perspective that deserves to be heard.

**Paper Summary:**

The paper argues that "integrating explicit memory is instrumental in advancing current AI towards Human-Level AI (HLAI)" [L014-016], claiming that the statistical learning is insufficient to incorporate symbolic reasoning and we need a form of explicit memory to supplement the implicit memory in LLMs. Most of the paper is spent describing the forms of memory in humans, while the rest of the paper discusses how LLMs don't have all the forms of memory that humans have.

**Position:**

Yes

**Position In Title:**

Yes

**Related Work:**

3

**Strengths And Weaknesses:**

### Strengths
- S1: **Gentle and thorough introduction to biological memory.** The paper provides an accessible overview to different kinds of memory (e.g., episodic vs. semantic, explicit vs. implicit) in biological systems. The AI community would benefit from understanding these well-studied phenomena in discussing the memory capabilities of large models.
- S2: **Tangible proposed implementation of the memory system.** In Section 6, the paper actual dives into how such explicit memory would be possible to encode in LLMs. I appreciate this formal aspect of the position paper, but it would need to be backed up by strong experimentation to compare to existing techniques.

### Weaknesses
- W1: **Questionable premises for the argument.** The argument hinges on the fact that current LLMs are incredibly limited and struggle to plan and do logical reasoning, and can't store and use information dynamically. However, current modeling paradigms can use tools across multiple agents, which largely solve these problems. LLMs are now capable of logical reasoning that surpasses even that of expert humans. The fact that the rest of the paper rests on this debatable assumption lowers the potential impact of the paper. Explicit memory already exists in markdown files in filesystems accessible to agents, IMO no need for "hippocampal style explicit memory".
- W2: **Overwhelming Table 1 term overload.** Table 1 is a term dump and hard to understand in context of the paper's argument
- W3: **Section 3 is a laundry list, not a story.** Section 3 lists a bunch of different ways to train an LLM, but it is unclear how those different aspects are connected to the argument of the paper.
- W4: **Unconvincing argument for metacognition.** When reading CoT in modern LLM systems, it becomes evident that LLMs are capable of reflecting on what it knows and doesn't know, in at least as great an extent as an average human is capable of knowing their own limitations.

### Conclusion
The paper is well written, but the argument that connects why biological models of memory are necessary for LLMs is weak. Indeed, there seems to be no real debate that "memory is necessary for HLAI", and the paper fails to convince that a "Hippocampal-style explicit memory" is necsessary for AI. Modern LLM systems are already capable of explicit memory by simply allowing models to write and read from scratchpads in accessible filesystem memory. Why is this insufficient?

This paper is better positioned as a good "glossary" to reference when discussing memory in AI systems.

**Support:**

2

---

> ### Author Rebuttal · Authors · 2026-03-30
>
> We would like to appreciate the reviewer for taking the time to review our research.
>
> **W1.** We agree that current LLMs show remarkable capabilities in certain well-defined tasks. However, to assess the gap towards Human-Level AI, it is critical to focus on the weakest aspects of current systems when compared to human cognition. In this regard, higher-order abilities such as metacognition, executive function, and mental simulation still exhibit a substantial gap to humans, and these gaps remain largely unresolved despite recent progress.
>
> Second, regarding mathematical reasoning, we acknowledge that LLMs achieve strong performance on many math benchmarks. However, this performance should be interpreted with caution. Humans are typically able to apply newly learned concepts to solve novel problems with minimal exposure, whereas LLMs generally require extensive training over large distributions of similar problems. Recent studies continue to show that LLMs struggle with problems that involve unfamiliar formulations or require flexible abstraction beyond their training distribution [1,2,3]. This suggests that high performance on known problem types does not necessarily imply the acquisition of human-like logical reasoning ability.
>
> [1] Sun, Y., et al. (2025). OMEGA: Can LLMs Reason Outside the Box in Math? Evaluating Exploratory, Compositional, and Transformative Generalization. *39th NeurIPS D&B Track*.
>
> [2] Zhou, Z., et al. (2024). Can Language Models Perform Robust Reasoning in Chain-of-thought Prompting with Noisy Rationales? *38th NeurIPS*.
>
> [3] Joishy, A., et al. (2026). Flying Pigs, FaR and Beyond: Evaluating LLM Reasoning in Counterfactual Worlds. *arXiv.*
>
> ---
>
> **W2.** We will streamline Table 1 by reducing the number of rows to more clearly and intuitively highlight our core arguments.
>
> ----
>
> **W3.** The intention behind Section 3 was not simply to enumerate various LLM training methods. Rather, we aimed to systematically demonstrate from multiple angles that the underlying mechanisms of current LLMs closely parallel the basal ganglia-based implicit memory system.
>
> We show that each of the technical characteristics, like gradual learning and dense coding, aligns directly with the biological properties of human implicit memory. This alignment is an essential foundational build-up for the core argument of our paper, providing the mechanistic evidence for our claim that current LLMs struggle with higher-order cognitive functions because their fundamental architecture remains trapped in an implicit memory system, lacking the mechanisms required for explicit memory.
>
> ---
>
> **W4.** It is true that recent LLMs show excellent performance in self-correction and confidence calibration with CoT, and this appears to operate similarly to human metacognition. However, according to recent studies, these capabilities can be distinguished from metacognition.
>
> Turpin et al. [1] report that the self-correction or uncertainty shown by LLMs is not a true reflection of their internal knowledge state, but rather a pattern learned through multitask learning and RLHF processes. In other words, when a model outputs "I do not know this," it is not metacognitive awareness, but rather it might be triggering self-validation in the middle of generation to receive higher rewards.
>
> According to Kadavath et al. [2], although they reported the self-evaluation capabilities of LLMs, they observed a limitation that this is not domain general but rather a restricted prediction that only operates in a specific trained domain. This suggests that the capability of the model is another form of probabilistic prediction depending on the training distribution.
>
> Similarly, Huang et al. [3] demonstrated fundamental limits to LLMs independently correcting logical errors in their responses using only their intrinsic capabilities, without the intervention of ground truth prompts or tools.
>
> In conclusion, although the uncertainty prediction and self-correction of LLMs have improved, these abilities are more accurately interpreted as multi-task learning where the model estimates the accuracy of generated text and utilizes that result as context. This mechanism has limitations in truly tracking the sources and boundaries of knowledge and in performing consistent self-objectification in unfamiliar environments. In particular, considering reports that issues such as continuously occurring hallucinations and the tendency to provide contradictory answers depending on the context are still common in long contexts or with uncommon inputs, it seems difficult to conclude that the model has acquired true metacognition.
>
> [1] Turpin, M., et al. (2023). Language Models Don’t Always Say What They Think: Unfaithful Explanations in Chain-of-Thought Prompting. *37th NeurIPS*.
>
> [2] Kadavath, S., et al. (2022). Language Models (Mostly) Know What They Know. *arXiv.*
>
> [3] Huang, J., et al. (2023). Large Language Models Cannot Self-Correct Reasoning Yet. *12th ICLR*.

---

> > ### Author Rebuttal · Reviewer_aFMf · 2026-04-02
> >
> > I thank the authors' for their response. I remain unconvinced by their argument that the memory described in this paper is necessary for future foundation models. However, it seems that the authors are capable of addressing and rewriting my concerns about presentation and story cohesion.
> >
> > With the assumption that the authors address my presentation concerns, I bump my score to a 4. Perhaps another reader will find the argument more convincing, and I do believe that the content of the paper is a perspective that deserves to be heard.

---

### Official Review · Reviewer_cNoq · 2026-03-17

**Significance:** 3
**Argument Clarity:** 2
**Rating:** 3
**Confidence:** 4

**Questions:**

My overall assessment is that this work requires substantial revision but deserves eventually a publication in a future venue. There is a great potential, but more care needs to be put into the position paper formatting guidelines.

**Alternative Views Section:**

Yes

**Compliance With Llm Reviewing Policy A Conservative:**

Affirmed.

**Discussion Potential:**

3

**Final Justification:**

As noted in my review, I believe substantial changes need to be made. Authors already committed to making them, which is why I increased my score. I do not oppose an acceptance, but in my opinion, the *submitted* work does not meet the bar for publication in ICML.

**Paper Summary:**

The position paper first performs a literature review on explicit vs implicit memory based on primarily correlational studies from old human neuroscience literature. Then, the authors argue that LLMs lack explicit memory, which is needed for AGI/HLAI.

**Position:**

Yes

**Position In Title:**

Yes

**Related Work:**

1

**Strengths And Weaknesses:**

Strengths:

- The gap in the literature this position paper aims to fill is solid. As authors state, LLMs are indeed having trouble with memory capabilities.

- The framing of the memory problem in terms of explicit and implicit memories, if done correctly, is a strong conceptual contribution. Though, see also weaknesses.

- Sections 5 and 6 seem very interesting and deserves eventual publication, though remains underdeveloped for a publication quality work as discussed below.

Weaknesses:

- One major weakness, especially for a neuroscientist, is the lack of rigor in the language used to describe correlational results. Most cited literature in neuroscience part is from old studies whose evidence is often not causal. But, the language is somewhat too causal and direct, which risks offending a great deal of neuroscientists. Regardless, rigor in scientific writing is the main point of a position paper, as no new result is really presented... It should be crystal clear what statements are supported by causal evidence vs which ones by correlational evidence, and whenever applicable, evidence should be spelled out. In a position paper, that is of utmost importance to incite a well-informed debate.

- The core contributions are underdeveloped. There is no reason to perform a literature review for 6 pages while leaving most of the exciting "positions" crammed in two pages. Sections 5 and 6 are quite exciting, but are rushed. Relatedly, no citation is provided to support any of the claims presented here. It is understood that position papers are supposed to go into a particular position and introduce new ideas, but they should still be anchored to existing ones to provide enough evidence that these ideas are worth pursuing. As they stand, Sections 5 and 6 have minimal evidence *presented in these sections* for their future relevance/feasibility.

- Relatedly, the paper relies heavily on partially outdated citations, as opposed to reviewing recent studies that have provided further evidence in our understanding of memory. Considering that about 6 out of 8 pages of the manuscript is effectively literature review, this is a major weakness.

**Support:**

2

---

> ### Author Rebuttal · Authors · 2026-03-30
>
> We would like to appreciate the reviewer for taking the time to review our research.
>
> **W1.** As you noted, distinguishing between causation and correlation is critically important in neuroscience. We have taken care to address this issue, but we will examine it more thoroughly again. However, we would like to emphasize that the central premise of our paper, that the hippocampus-based explicit memory system is a necessary condition for higher cognitive functions such as future planning and reasoning, is not based on mere correlation but on well-established causal evidence accumulated over decades in neuroscience.
>
> The relationship between explicit memory and executive function discussed in the manuscript has been cross-validated through studies of patients with bilateral hippocampal lesions, such as Patient H.M., as well as through various animal deficit models [1, 2]. The fact that loss of hippocampal function leads to the absence of explicit memory, which in turn directly results in impairments in goal-directed behavior and the ability to simulate future scenarios, provides clear evidence of a causal deficit. In other words, the loss of a specific brain structure directly causes the breakdown of particular cognitive abilities [3].
>
> Another important issue concerns causal evidence at the level of specific subregions and detailed mechanisms within the hippocampus. In this regard, many studies have demonstrated causality at both the molecular and circuit levels. For example, Goshen et al. [4] used optogenetics to precisely and temporally inhibit the CA1 subregion of the hippocampus in real time, showing that this circuit plays an essential causal role not only in recent memory but also in the retrieval of long-term (remote) memories. In addition, Ramirez et al. [5] demonstrated that optogenetic activation of a specific ensemble of engram cells in the dentate gyrus is sufficient to reconstruct a prior contextual memory and even generate false memories. This provides direct evidence that activity in a specific subcircuit can serve as a sufficient condition for memory retrieval.
>
> [1] SCOVILLE, W. B., & MILNER, B. (1957). Loss of recent memory after bilateral hippocampal lesions. Journal of neurology, neurosurgery, and psychiatry, 20(1), 11–21.
>
> [2] Squire, L. R., & Zola-Morgan, S. (1991). The medial temporal lobe memory system. *Science (New York, N.Y.)*, *253*(5026), 1380–1386.
>
> [3] Hassabis, D., & Maguire, E. A. (2007). Deconstructing episodic memory with construction. *Trends in cognitive sciences*, *11*(7), 299–306.
>
> [4] Goshen, I., Brodsky, M., Prakash, R., Wallace, J., Gradinaru, V., Ramakrishnan, C., & Deisseroth, K. (2011). Dynamics of retrieval strategies for remote memories. *Cell*, *147*(3), 678–689.
>
> [5] Ramirez, S., Liu, X., Lin, P. A., Suh, J., Pignatelli, M., Redondo, R. L., Ryan, T. J., & Tonegawa, S. (2013). Creating a false memory in the hippocampus. *Science*, *341*(6144), 387–391.
>
> ---
>
> **W2.** We sincerely appreciate your positive feedback describing the ideas in Sections 5 and 6 as exciting. In response, we aim to clarify our intentions and outline how we will improve the paper.
>
> The core contribution of this study is to provide one reasonable perspective and interpretation for understanding the fundamental nature of LLM intelligence, supported by valid evidence and logical reasoning. Building upon this, our goal was to offer valuable insights for addressing current challenges in the field.
>
> While our argument in Section 5 regarding the necessity of an explicit memory is a crucial aspect of our work, our primary differentiator from prior research lies in Sections 2 through 4. In these earlier sections, we directly analyze and dissect the low-level mechanisms of both LLM functions and human cognitive abilities to draw novel insights.
>
> In fact, the novelty of our assertion in Section 5 stems from the foundations laid in the preceding sections. Specifically, we establish that LLMs share a structural similarity with implicit memory systems, that their current limitations are deeply tied to explicit memory functions, and that these two systems operate using fundamentally different mechanisms. Following this logical progression, Section 5 serves as a natural derivation of our conclusions.
>
> Regarding your point about the relationship with existing methodologies, we have included a comparative analysis of MemoryLLM, Larimar, Memoria, and Titans in Appendix A: Recent Progress. In this section, we clearly detail the contributions of these prior studies as explicit memory systems, alongside their limitations when viewed from the perspective of our paper.
>
> ---
>
> **W3.** We made every effort to rely on studies whose findings, although published in the past, continue to be recognized as valid today. Nevertheless, if any of the evidence we have cited is found to contradict conclusions established by more recent research, we would appreciate your guidance and will examine the matter carefully.

---

> > ### Author Rebuttal · Reviewer_cNoq · 2026-04-02
> >
> > Thanks to the authors for the rebuttal. My concerns may have been partially misunderstood, as the answers provided by the authors seem somewhat tangential to the concerns I raised. Please allow me to elaborate:
> >
> > 1) My concern is with the language. In many sentences referring to literature, the *nature* of the evidence is not clearly stated. When referring to evidence from a biology work, it is often desirable to explicitly lay out what the evidence is and make it clear if that is correlational or causal. I did not disagree with the fact that causal evidence exists here. For instance "while semantic memory is essential for knowledge-based reasoning, such as language processing and categorization (Binder & Desai, 2011)." What evidence was provided by Binder & Desai, 2011 to make this argument? All I asked was a structured rewrite of sentences of this form. I believe more care on this matter would increase the impact of this position paper with a major target audience: the neuroscientists.
> >
> > 2) Once again, my concern here is not about the content but style. The build-up towards Section 5 cites many works and has interesting arguments, agreed. But, since they are written in a review style, it is not always easy to dissect which perspectives are new and which ones are established in the field. Hence, my comment about the prior sections reading like a review paper. Afterwards, when assertions are made in Section 5 without clear citations connecting back to earlier sections, it is left to the reader to make these connections by going back and forth. As noted in my original review, "As they stand, Sections 5 and 6 have minimal evidence presented in these sections for their future relevance/feasibility." I do not see a clear effort in addressing this concern, rather the response seems to point me towards a different section. I simply think that the current organization places unnecessary burden on the reader to dissect the new from the old ideas, and should undergo major revisions.
> >
> > 3) Alas, the field's standards for what we call evidence changes over the years. Many of these experiments are repeated later on, some of which are cited in the rebuttal response but not in the original paper. All I requested was to supplement more modern citations, no reason to take out the early ones. Especially for works in biology, modern evidence is almost always available and citing them beyond textbook examples (e.g., patient HM here...) would only increase the impact of the position paper. The response by the authors here seems to miss or sidesteps this point entirely and decreases my confidence in recommending an acceptance without further reviewer guidance.
> >
> > I am happy to hear back from the authors. I will adjust my score in the end based on the final response, i.e., whether the response will provide convincing evidence that authors are willing to address these concerns on their own in the case of an acceptance. These points may seem inconsequential, or less important, compared to the main contributions of the work; but they would be to a neuroscience crowd that I believe this position should also excite! Point 2, in particular, is important for a clear assignment of ideas with respect to the "studies whose findings, although published in the past, continue to be recognized as valid today."

---

### Decision · Program_Chairs · 2026-04-30

**Decision:**

Accept (regular)

**Comment:**

This paper provides good review and comparison on implicit and explicit memories, with interesting propositions to incorporate explicit memories. The review process focused on the distinction between mimicry of memory and its architectural necessity:
•	A significant insight derived from the discussion was the "Executive Function Gap." While humans can immediately follow a new rule (e.g., "From now on, replace all 'a's with 'e's"), LLMs struggle to maintain consistency throughout a response. This was used as evidence that models lack a "working memory" grounded in explicit logical structures.
•	Reviewers questioned how the model would know when to consult its explicit memory as a form of metacognitive monitoring. The authors clarified in their rebuttal that an integrated hippocampal system would allow the agent to "know what it doesn't know", hence performing a confidence check against stored facts before generating a response.
•	A critical point raised during the rebuttal phase was whether explicit memories should eventually become implicit (consolidated into weights). The authors updated their position to suggest that HLAI requires a constant bi-directional flow between the "fast" hippocampal system and the "slow" cortical weight system.